# Maxwell Nanofluids: FEM Simulation of the Effects of Suction/Injection on the Dynamics of Rotatory Fluid Subjected to Bioconvection, Lorentz, and Coriolis Forces

**DOI:** 10.3390/nano12193453

**Published:** 2022-10-02

**Authors:** Liaqat Ali, Abdul Manan, Bagh Ali

**Affiliations:** 1School of Sciences, Xi’an Technological University, Xi’an 710021, China; 2Department of Physics and Mathematics, Faculty of Sciences, Superior University, Lahore 54000, Pakistan; 3Department of Computer Science and Information Technology, Faculty of Sciences, Superior University, Lahore 54000, Pakistan

**Keywords:** Maxwell nanofluid, finite element analysis, suction/injection, grid independence analysis, Coriolis force

## Abstract

In this study, the relevance of Lorentz and Coriolis forces on the kinetics of gyratory Maxwell nanofluids flowing against a continually stretched surface is discussed. Gyrotactic microbes are incorporated to prevent the bioconvection of small particles and to improve consistency. The nanoparticles are considered due to their valuable properties and ability to enhance thermal dissipation, which is important in heating systems, advanced technology, microelectronics, and other areas. The main objective of the analysis is to enhance the rate of heat transfer. An adequate similarity transformation is used to convert the primary partial differential equations into non-linear dimensionless ordinary differential equations. The resulting system of equations is solved using the finite element method (FEM). The increasing effects of the Lorentz and Coriolis forces induce the velocities to moderate, whereas the concentration and temperature profiles exhibit the contrary tendency. It is observed that the size and thickness of the fluid layers in the axial position increase as the time factor increases, while the viscidity of the momentum fluid layers in the transverse path decreases as the time factor decreases. The intensity, temperature, and velocity variances for the suction scenario are more prominent than those for the injection scenario, but there is an opposite pattern for the physical quantities. The research findings are of value in areas such as elastomers, mineral productivity, paper-making, biosensors, and biofuels.

## 1. Introduction

The heat and mass transfer analysis of the non-Newtonian hydrodynamic boundary layer flow phenomenon has attracted the interest of many researchers due to the enormous number of potential applications in engineering and industry. The well-known Newtonian liquids (liquids with a sequential strain-stress correlation) basic theory is incapable of elucidating the fluids’ internal microstructure. Non-Newtonian liquids (liquids with a sequential strain-stress correlation) include quince paste, animal blood, cement sludges, esoteric lubricating oils, effluent slurry, and liquids containing synthetic polymer additives. One such rate-type non-Newtonian fluid model is called the Maxwell nanofluid model which predicts the stress relaxation time. The extensive choice of methodological and engineering applications associated with Maxwell nanofluids, such as biochemical, gasoline, polymer, and nutrition release, has motivated many investigators to scrutinize the features of Maxwell nanofluids with respect to numerous geometrical and substantial limitations. The convective Maxwell hybrid nanofluid stream in a sturdy channel was studied using the Laplace transform strategy [1], whereby the authors developed a solution to dynamical problems involving Maxwell fluid fractionally. The Caputo fractional differential function was used for the energy dissipation assessment of hydromagnetic Maxwell nanofluid flow over an elongating penetrable surface with Dufour and Soret ramifications. Jawad et al. [2] used HAM to procure estimated analytical results. Jamshed [3] exploited the Keller box technique (KBT) to investigate the fluidity of an mhd Maxwell nanofluid over a non-linearly elongating sheet in terms of viscous dissipation and entropy propagation. Ali et al. [4] investigated buoyant, induced transitory bio-convective Maxwell nanoliquid spinning three-dimensional flows over the Riga surface for chemically reactive and activating energy using a finite element stratagem. Dulal et al. [5] presented results of a study on mhd radiative heat transfer of nanofluids induced by a plate through a porous medium with chemical reaction. Very recently, various authors have explored boundary layer Maxwell nanofluid flow past a different geometric environment. These include Ahmed et al. [6], who reported on mixed convective 3D flow over a vertical stretching cylinder with a shooting technique, Gopinath et al. [7] who explored convective-radiative boundary layer flow of nanofluids with viscous-Ohmic dissipation, Bilal et al. [8] who presented the significance of the Coriolis force on the dynamics of the Carreau–Yasuda rotating nanofluid subject to gyrotactic microorganisms, Ahmed [9] who investigated the effect of a heat source on the stagnation point fluid flow via an elongating revolving plate using a numerical approach, M. Bilal [10] who used the HAM technique to investigate chemically reactive impacts on magnetised nanofluid flow over a rotary pinecone, Amirsom et al. [11] who estimated the influence of bioconvection on three-dimensional nanofluid flow induced by a bi-axial stretching sheet, Prabhavathi et al. [12] who used FEM to investigate CNT nanofluid flow through a cone with thermal slip scenarios, Zohra et al. [13] who used mhd micropolar fluid bio-nanoconvective Naiver slip flow in a stretchable horizontal channel, and Gopinath et al. [14] who reported on diffusive mhd nanofluid flow past a non-linear stretching/shrinking sheet with viscous-Ohmic dissipation and thermal radiation.

Nanofluids are fluids that incorporate an appropriate distribution of metal and metallic nanoparticles at the nano size and are engineered to perform specific functions [15,16,17,18]. The literature suggests that the presence of nanoparticles in a base fluid has a significant impact on the thermophysical properties of the fluid, particularly those fluids with inadequate permittivity characteristics based on theoretical and experimental investigations [19,20,21]. Applications in virtually every field of engineering and science relating to convective nanofluid heat transfer flow have stimulated the interest of many scientists and engineers. These include the use of diamond and silica nanoparticles to enhance the electrical characteristics of lubricants, the use of liquids containing nanoparticles to absorb sunlight in solar panels, and exploitation of the antimicrobial properties of zinc and titanium oxide particles for biomedical engineering applications, such as drug delivery and pharmacological treatment [22,23,24,25].

The bioconvection phenomenon occurs as a result of the existence of a density gradient in the flow field. Consequently, the movement of particles at the macroscopic level enhances the density stratification of the base liquid in one direction. The presence of gyrotactic microorganisms in nanofluid flow has attracted the interest of many researchers due to their potential application in relation to enzyme function, bio-sensors, biotechnology, drug delivery, and biofuels. These applications have motivated researchers to undertake numerical studies on bioconvective nanofluid flow with microorganisms in different flow field geometries. Chu et al. [26] investigated bioconvective Maxwell nanoliquid flow using a reversible, regularly pivoting sheet in the presence of non-linear radiative and heat emitter influences using a homotopy analysis method. Sreedevi et al. [27] investigated the influence of Brownian motion and thermophoresis on Maxwell three-dimensional nanofluid flow over a stretching sheet with thermal radiation. Rao et al. [28] explored bioconvection in conventional reactive nanoliquid flow over a vertical cone with gyrotactic microorganisms embedded in a permeable medium. Awais Ali et al. [29], using an Adams–Bash strategy (ABS), statistically explored the Lie group, to investigate bio-convective nanoliquid supporting and opposing flow with motile microorganisms. To determine the Arrhenius activation energy of bio-convective nanoliquid flow through a stretchable surface, Paluru [30] undertook a heat and mass transfer analysis of MWCNT-kerosene nanofluid flow over a wedge with thermal radiation. Transient bio-convective Carreau nanofluid flow with gyrotactic microorganisms past a horizontal slender stretching sheet was considered by Elayarani et al. [31] to investigate heat and mass transfer effects in the presence of thermal radiation, multi-slip conditions, and magnetic fields by employing the ANFIS (adaptive neuro-fuzzy inference system) model. Bagh et al. [32] reported on the g-jitter impact on magnetohydrodynamic non-Newtonian fluid over an inclined surface by applying a finite element simulation. Umar et al. [33] investigated the optimized Cattaneo–Christov heat and mass transference flow of bio-convective Carreau nanofluid with microorganisms, influenced by a longitudinal straining cartridge with convective limitations. Al-Hussain [34] developed an analytical model based on the Cattaneo–Christov transit law for a bio-convective magnetic nanofluid stream via a whirling cone immersed in an asymmetric penetrable surface in the context of cross-diffusion, Navier-slip, and Stefan blowing effects.

The careful review of the literature detailed above shows that little attention has been paid to the self-motile denitrifying microbes contained in Maxwell nanofluid spinning flows through an elongating sheet under an externally applied magnetic field. To the best of our knowledge, none of the studies cited has considered the interpreted problem. Consequently, the main objective of this study was to explore the mass and heat transfer impacts on transitory hydromagnetic Maxwell spinning nanofluid 3D radiative flow comprising microbes and suction/injection processes. Many authors [35,36,37] have examined mhd nanofluid flow using different numerical techniques. Here, the flow-governing associated non-linear PDEs are computed using a finite volume technique [38,39] by adopting a weighted residual strategy. The varied flow field properties for a variety of substantial factors are explained and illustrated graphically. The computing results generated using Matlab source code were validated against previous studies and determined to show acceptable consistency. The values of the friction factor, Nusselt, and Sherwood numbers are simulated and addressed in tabulated form. The computational evaluation can be used for gasoline, polymers, precise nutrition release, engine lubricants, paint rheology, biosensors, medicine delivery, and biofuels.

### Research Queries

The following research questions are addressed in this study:What effects do relaxation of the Deborah number, the Coriolis effect, and an applied magnetic field force have on the hydrodynamics of heat flux, fluid viscosity, and concentration level variances using injection/suction?What impact do Brownian motion and thermophoresis have on heat and mass transfer rates and the skin friction factor for suction/injection?How do Brownian motion, the relaxation Deborah number, and time-dependent factors impact on the temperature profile?What is the bioconvection impact on the motile dispersal function with suction/injection?

## 2. Mathematical Geometry

The transitory magneto-hydrodynamic 3D rotational flow of Maxwell nanofluid over a bidirectional elongating surface is investigated. Figure 1 depicts the fluid dynamic layout and coordinate structure of the articulated problem, with the flow, constrained to z≥0. With a rotational consistent velocity Ω, the nanofluid flow rotates around the z-axis. When z=t=0.0, the sheet is extended along the x-axis having $u_{w}=\tilde{a}x$ velocity. In the axial direction, a static and uniform magnetic field of magnitude $B_{0}$ is implemented. An induced magnetic Reynolds number leads to a reduced magnetic field, which results in minimal Hall current and Ohmic inefficiency [40]. To avoid causing sedimentation, gyrotactic microbes are utilized to maintain convectional stability. The external temperature and intensity are signified by $T_{\infty }$, and $C_{\infty }$, $N_{\infty }$, respectively, while the temperature and intensity at the surface are represented by $T_{w}$, and $C_{w}$, $N_{w}$, respectively. For the current elaborated problem, $V=(u_{1}\left(x,y,z\right),u_{2}\left(x,y,z\right),u_{3}\left(x,y,z\right))$ is assumed to be the velocity field. The equations of mass conservation, linear moments, temperature, and concentrations are formulated as a result of the preceding assertions [41,42,43]:(1)$$\begin{array}{c} \partial_{x}u_{1}+\partial_{y}u_{2}+\partial_{z}u_{3}=0 \end{array}$$(2)$$\begin{array}{c} \rho_{nf}\left(\partial_{t}u_{1}+u_{1}\partial_{x}u_{1}+u_{2}\partial_{y}u_{1}+u_{3}\partial_{z}u_{1}-2\Omega u_{2}+\lambda_{1}\varrho_{u_{1}}\right)=-\partial_{x}p+\mu_{nf}\partial_{zz}u_{1}-\sigma_{n_{f}}B_{0}^{2}u_{1} \end{array}$$(3)$$\begin{array}{c} \rho_{nf}\left(\partial_{t}u_{2}+u_{1}\partial_{x}u_{2}+u_{2}\partial_{y}u_{2}+u_{3}\partial_{z}u_{2}-2\Omega u_{1}+\lambda_{1}\varrho_{u_{2}}\right)=-\partial_{y}p+\mu_{nf}\partial_{zz}u_{2}-\sigma_{n_{f}}B_{0}^{2}u_{2} \end{array}$$
(4)$$\begin{array}{c} \rho_{nf}\left(\partial_{t}u_{3}+u_{1}\partial_{x}u_{3}+u_{2}\partial_{y}u_{3}+u_{3}\partial_{z}u_{3}\right)=-\partial_{z}p+\mu_{nf}\partial_{zz}u_{3} \end{array}$$
(5)$$\begin{array}{c} \partial_{t}T+u_{1}\partial_{x}T+u_{2}\partial_{y}T+u_{3}\partial_{z}T=\alpha_{n_{f}}\partial_{zz}T+\tau^{*}\left\{D_{b}\partial_{z}C\partial_{z}T+\frac{D_{T}}{T_{\infty }}{\left(\partial_{z}T\right)}^{2}\right\} \end{array}$$
(6)$$\begin{array}{c} \partial_{t}C+u_{1}\partial_{x}C+u_{2}\partial_{y}C+u_{3}\partial_{z}C=D_{b}\partial_{zz}C+\frac{D_{T}}{T_{\infty }}\partial_{zz}T \end{array}$$
(7)$$\begin{array}{c} \partial_{t}N+u_{1}\partial_{x}N+u_{2}\partial_{y}N+u_{3}\partial_{z}N+\frac{bWc}{\left(C_{w}-C_{\infty }\right)}\left[\partial_{z}\left(N\partial_{z}C\right)\right]=D_{m}\partial_{zz}N \end{array}$$
where,
$$\begin{array}{c} \varrho_{u_{1}}=\{u_{1}^{2}\partial_{xx}u_{1}+u_{2}^{2}\partial_{yy}u_{1}+u_{3}^{2}\partial_{zz}u_{1}+2u_{1}u_{2}\partial_{xy}u_{1}+2u_{2}u_{3}\partial_{yz}u_{1}+2u_{1}u_{3}\partial_{xz}u_{1}\\ -2\Omega \left(u_{1}\partial_{x}u_{2}+u_{2}\partial_{y}u_{2}+u_{3}\partial_{z}u_{2}\right)+2\Omega \left(u_{2}\partial_{x}u_{1}-u_{1}\partial_{y}u_{1}\right)\},\\ \varrho_{u_{2}}=\{u_{1}^{2}\partial_{xx}u_{2}+u_{2}^{2}\partial_{yy}u_{2}+u_{3}^{2}\partial_{zz}u_{2}+2u_{1}u_{2}\partial_{xy}u_{2}+2u_{2}u_{3}\partial_{yz}u_{2}+2u_{1}u_{3}\partial_{xz}u_{2}\\ -2\Omega \left(u_{1}\partial_{x}u_{1}+u_{2}\partial_{y}u_{1}+u_{3}\partial_{z}u_{1}\right)+2\Omega \left(u_{2}\partial_{x}u_{2}-u_{1}\partial_{y}u_{2}\right)\} \end{array}$$

Here, (C,N,T) represent the nanoparticle density, micro-organism concentration, and fluid temperature, ($D_{b},D_{T},D_{m}$) are the Brownian motion, thermophoresis and microorganism diffusion; ($\lambda_{1},\rho_{n_{f}},\mu_{n_{f}},\alpha_{n_{f}}$) are, respectively, the relaxation time, density, dynamic viscosity, and thermal diffusivity of the nanofluid. The boundary conditions are [44,45]:(8)$$\begin{array}{c} t<0:u_{1}=0,\;u_{2}=0,\;u_{3}=0,\;C=\left(C_{\infty }\right),\;N=\left(N_{\infty }\right),\;T=\left(T_{\infty }\right) \end{array}$$(9)$$\begin{array}{c} t\geq 0:u_{1}=a\left(x\right),\;u_{3}=-w_{0},\;u_{2}=0,\;C=\left(C_{w}\right),\;N=\left(N_{w}\right),\;T=\left(T_{w}\right),\;when\;z=0 \end{array}$$(10)$$\begin{array}{c} t\geq 0:u_{1}\to 0,\;u_{2}\to 0,\;C\to C_{\infty },\;N\to N_{\infty },\;T\to T_{\infty },\;when\;z\to \infty . \end{array}$$

The following similarity transforms are used to alleviate the complexity of the articulated problem as [41,44]:(11)$$\begin{array}{c} \left.\begin{array}{c} \Gamma =\tilde{a}t,\;u_{1}=\tilde{a}x\frac{\partial F(\zeta ,\eta )}{\partial \eta },\;u_{2}=\tilde{a}xG\left(\zeta ,\eta \right),\;u_{3}=-\sqrt{\tilde{a}\nu \zeta }F\left(\zeta ,\eta \right),\;\;\zeta =1-e^{-\Gamma },\eta =\sqrt{\frac{\tilde{a}xz^{2}}{\zeta \nu }}\\ C=\left(C_{w}-C_{\infty }\right)\Phi \left(\zeta ,\eta \right)+C_{\infty },\;N=\left(N_{w}-N_{\infty }\right)\chi \left(\zeta ,\eta \right)+N_{\infty },\;T=\left(T_{w}-T_{\infty }\right)\theta \left(\zeta ,\eta \right)+T_{\infty } \end{array}\right\} \end{array}$$

In the context of Equation (Equation 11), Equation (Equation 1) is justified, and Equations (2)–(10) are transmuted into the non-linear PDEs illustrated below in (ζ,η) form:(12)$$\begin{array}{c} F^{\prime \prime \prime }+\frac{\eta }{2}F^{\prime \prime }-\frac{\zeta \eta }{2}F^{\prime \prime }+\zeta \left\{FF^{\prime \prime }-F^{{}^{\prime }2}-M^{2}F^{\prime }+2\lambda G+\beta \varsigma_{u_{1}}\right\}-\zeta \left(1-\zeta \right)\frac{\partial F^{\prime }}{\partial \zeta }=0 \end{array}$$(13)$$\begin{array}{c} G^{\prime \prime }+\frac{\eta }{2}G^{\prime }-\frac{\zeta \eta }{2}G^{\prime }+\zeta \left\{FG^{\prime }-2\lambda F^{\prime }-M^{2}G-F^{\prime }G+\beta \varsigma_{u_{2}}\right\}-\zeta \left(1-\zeta \right)\frac{\partial G}{\partial \zeta }=0 \end{array}$$(14)$$\begin{array}{c} \theta^{\prime \prime }-\frac{\eta }{2}\left(\zeta -1\right)Pr\theta^{\prime }+\zeta PrF\theta^{\prime }+N_{b}Pr\theta \Phi +N_{t}Pr\theta^{{}^{\prime }2}-\zeta \left(1-\zeta \right)Pr\frac{\partial \theta }{\partial \zeta }=0 \end{array}$$(15)$$\begin{array}{c} \Phi^{\prime \prime }+0.5\eta Le\left(1-\zeta \right)\Phi^{\prime }+Le\zeta F\Phi^{\prime }+N_{t}N_{b}^{-1}\theta^{\prime \prime }-\zeta \left(1-\zeta \right)Le\frac{\partial \Phi }{\partial \zeta }=0 \end{array}$$(16)$$\begin{array}{c} \chi^{\prime \prime }+\frac{Lb}{2}\left(1-\zeta \right)Lb\chi^{\prime }+\zeta LbF\chi^{\prime }-Pe\Phi^{\prime \prime }\left(\delta_{1}+\chi \right)+Pe\chi^{\prime }\Phi^{\prime }=Lb\zeta \left(1-\zeta \right)\frac{\partial \chi }{\partial \zeta } \end{array}$$(17)$$\begin{array}{c} \left.\begin{array}{c} F\left(\zeta ,\eta \right)=\Gamma ,\;F^{\prime }\left(\zeta ,\eta \right)=\theta \left(\zeta ,\eta \right)=1,\;G\left(\zeta ,\eta \right)=0,\Phi \left(\zeta ,\eta \right)=\chi \left(\zeta ,\eta \right)=1,\;\zeta \geq =0,\;when\;\eta =0\\ F^{\prime }\left(\zeta ,\eta \right)\to 0,\;\theta \left(\zeta ,\eta \right)\to 0,\;\;G\left(\zeta ,\eta \right)\to 0,\;\Phi \left(\zeta ,\eta \right)\to 0,\;\chi \left(\zeta ,\eta \right)\to 0,\;\zeta \geq 0,\;when\;\eta \to \infty \end{array}\right\} \end{array}$$
where $\varsigma_{u_{1}}=2FF^{\prime }F^{\prime \prime }-F^{2}F^{\prime \prime \prime }-2\lambda FG^{\prime }$, $\varsigma_{u_{2}}=2FF^{\prime }G^{\prime }-F^{2}G^{\prime \prime }-2\lambda F^{\prime 2}+2\lambda FF^{\prime \prime }-2\lambda G^{2}$, and primes (${}^{\prime },{}^{\prime \prime },{}^{\prime \prime \prime }$) denote the derivatives w.r.t (η). Here rotation, magnetized, Prandtl and Lewis numbers, Brownian factor, bioconvection Lewis number, thermophoresis, Peclet number, relaxation Deborah number, microorganism concentration difference, and suction/injection are, respectively, λ, *M*, Pr, Le, $N_{b}$, Lb, $N_{t}$, Pe, β, $\delta_{1}$, and Γ factors are described as:$$\begin{array}{c} \lambda =\frac{\Omega }{a},\;M=\sqrt{\frac{\sigma_{n_{f}}B_{o}^{2}}{\rho_{f}\tilde{a}}},\;\;Pr=\frac{\nu }{\alpha_{n_{f}}},\;Le=\frac{\nu }{D_{B}},\;Lb=\frac{\nu }{D_{m}},\;N_{b}=\tau \nu^{-1}D_{B}\left(C_{w}-C_{\infty }\right),\\ N_{t}=\frac{D_{T}\left(\tau T_{w}-\tau T_{\infty }\right)}{\nu T_{\infty }},\;\beta =\lambda_{1}a,\;Pe=\frac{bW_{c}}{D_{m}},\;\delta_{1}=\frac{N_{\infty }}{N_{w}-T_{\infty }},\;\Gamma =\frac{w_{0}}{\sqrt{\tilde{a}\nu \zeta }}. \end{array}$$

The physical quantities (Sherwood, Nusselt) numbers, and the coefficient of skin friction are expressed here as:(18)$$\begin{array}{c} Nu_{x}=\frac{xq_{w}}{\kappa (T_{w}-T_{\infty })},\;Shr=\frac{xq_{m}}{D_{B}\left(C_{w}-C_{\infty }\right)},\;C_{f_{x}}=\frac{\tau_{w}^{x}}{\rho u_{1}^{2}},\;C_{f_{y}}=\frac{\tau_{w}^{y}}{\rho u_{1}^{2}}. \end{array}$$
here, the skin friction tensors at the wall are represented as $\tau_{w}^{x}=\mu {\left(\frac{\partial u_{1}}{\partial z}\right)}_{z=0}$ (along the x-axis) and $\tau_{w}^{y}=\mu {\left(\frac{\partial u_{2}}{\partial z}\right)}_{z=0}$ (along the y-axis), the heat flux, and the mass at the surface is $qm=-D_{B}{\left(\frac{\partial C}{\partial z}\right)}_{z=0}$, and $qw=-\kappa {\left(\frac{\partial T}{\partial z}\right)}_{z=0}$. Taking Equation (Equation 11), we get: (19)$$\left\{\begin{array}{c} C_{f_{x}}{Re_{x}}^{1/2}=\frac{F^{\prime \prime }\left(0\right)}{\sqrt{\zeta }},C_{f_{y}}{Re_{x}}^{1/2}=\frac{G^{\prime }\left(0\right)}{\sqrt{\zeta }},\\ Nu_{x}{Re_{x}}^{1/2}=-\frac{\left[\theta^{\prime }\left(0\right)\right]}{\sqrt{\zeta }},Shr_{x}{Re_{x}}^{1/2}=-\frac{\left[\Phi^{\prime }\left(0\right)\right]}{\sqrt{\zeta }}. \end{array}\right.$$

## 3. Computational Procedure

The finite element analysis (FEA) is a computational approach for discovering numerical approximations to ODEs and PDEs with complicated boundary conditions. This is an efficient approach for resolving technological problems, especially those involving fluid diversities. This methodology represents an excellent numerical strategy for solving a variety of real-world problems, particularly for heat transfer via fluids and biomaterials [46]. Reddy [47,48] presents a layout of the Galerkin finite element methodology (GFEM), summarizing the main elements of this methodology. This methodology is an unparalleled computational methodology in the field of engineering, is valuable for evaluating integral governing equations incorporating fluid diversities, and is an extremely effective methodology for resolving numerous non-linear problems [49,50,51]. To evaluate the set of Equations (12)–(16) along the boundary condition (17), firstly, we assume:(20)$$\begin{array}{c} F^{\prime }=P \end{array}$$

The system of Equations (12)–(17) reduced as:(21)$$\begin{array}{c} P^{\prime \prime }-\frac{\eta }{2}\left(\zeta -1\right)P^{\prime }+\zeta \left(FP^{\prime }-P^{2}+2\lambda G-M^{2}P+\beta \left(2FPP^{\prime }-F^{2}P^{\prime \prime }-2\beta FG^{\prime }\right)\right)=\zeta \frac{\partial P}{\partial \zeta }-\zeta^{2}\frac{\partial P}{\partial \zeta }\\ G^{\prime \prime }+\frac{1}{2}\left(1-\zeta \right)\eta G^{\prime }+\zeta \left(FG^{\prime }+\beta \left(2FPG^{\prime }-F^{2}G^{\prime \prime }-2\lambda P^{2}+2\beta FP^{\prime }-2\beta G^{2}\right)\right) \end{array}$$
(22)$$\begin{array}{c} -PG-2\beta P=-\zeta^{2}\frac{\partial \left(G\right)}{\partial \zeta }+\zeta \frac{\partial \left(G\right)}{\partial \zeta } \end{array}$$
(23)$$\begin{array}{c} \theta^{\prime \prime }-\frac{\eta }{2}\left(\zeta -1\right)Pr\theta^{\prime }+Pr\zeta F\theta^{\prime }+Pr\theta^{\prime }\left(N_{b}\Phi^{\prime }+N_{t}\theta^{\prime }\right)=Pr\zeta \left(1-\zeta \right)\frac{\partial \theta }{\partial \zeta } \end{array}$$
(24)$$\begin{array}{c} \Phi^{\prime \prime }+\frac{\eta }{2}Le\left(1-\zeta \right)\Phi^{\prime }+Le\zeta F\Phi^{\prime }+N_{t}N_{b}^{-1}\theta^{\prime \prime }{}^{2}=\zeta \left(1-\zeta \right)Le\frac{\partial \Phi }{\partial \zeta } \end{array}$$
(25)$$\begin{array}{c} \chi^{\prime \prime }+\frac{Lb}{2}\left(1-\zeta \right)\eta \chi^{\prime }+\zeta LbF\chi^{\prime }-Pe\Phi^{\prime \prime }\left(\delta_{1}+\chi \right)+Pe\chi^{\prime }\Phi^{\prime }=Lb\zeta \left(1-\zeta \right)\frac{\partial \chi }{\partial \zeta } \end{array}$$
(26)$$\begin{array}{c} \left.\begin{array}{c} F(\zeta ,\eta )=\Gamma ,\;G(\zeta ,\eta )=0,\;P(\zeta ,0)=\theta (\zeta ,\eta )=\Phi (\zeta ,\eta )=\chi (\zeta ,\eta )=1,\;\zeta \geq =0,\;at\;\eta =0\\ P(\zeta ,\eta )\to 0,\;G(\zeta ,\eta )\to 0,\;\theta (\zeta ,\eta )\to 0,\;\Phi (\zeta ,\eta )\to 0,\;\chi (\zeta ,\eta )\to 0,\;\zeta \geq 0,\;as\;\eta \to \infty \end{array}\right\} \end{array}$$

For numerical calculation, the plate length has been specified as ζ=1.0 and the thickness as η=5.0. The Equations (20)–(25) have a variational form that can be represented as:(27)$$\begin{array}{c} \int_{\Omega_{e}}w_{f_{1}}\left\{F^{\prime }-P\right\}d\Omega_{e}=0\\ \int_{\Omega_{e}}w_{f_{2}}\{P^{\prime \prime }+\frac{1}{2}\left(1-\zeta \right)\eta P^{\prime }+\zeta \left(FP^{\prime }-P^{2}+2\lambda P-M^{2}P+\beta \left(2FPP^{\prime }-F^{2}P^{\prime \prime }-2\lambda FG^{\prime }\right)\right) \end{array}$$(28)$$\begin{array}{c} -\zeta \left(1-\zeta \right)\frac{\partial P}{\partial \zeta }\}d\Omega_{e}=0\\ \int_{\Omega_{e}}w_{f_{3}}\{G^{\prime \prime }+\frac{1}{2}\left(1-\zeta \right)\eta G^{\prime }+\zeta \left(FG^{\prime }+\beta \left(2FPG^{\prime }-F^{2}G^{\prime \prime }-2\lambda P^{2}+2\lambda FP^{\prime }-2\lambda G^{2}\right)\right) \end{array}$$(29)$$\begin{array}{c} -PG-2\lambda P+\left(\zeta -1\right)\zeta \frac{\partial \left(G\right)}{\partial \zeta }\}d\Omega_{e}=0 \end{array}$$(30)$$\begin{array}{c} \int_{\Omega_{e}}w_{f_{4}}\left\{\theta^{\prime \prime }+\frac{Pr}{2}\left(1-\zeta \right)\eta \theta^{\prime }+Pr\zeta F\theta^{\prime }+N_{b}Pr\theta^{\prime }\Phi^{\prime }+N_{t}Pr{\left(\theta^{\prime }\right)}^{2}-Pr\zeta \left(1-\zeta \right)\frac{\partial \theta }{\partial \zeta }\right\}d\Omega_{e}=0 \end{array}$$(31)$$\begin{array}{c} \int_{\Omega_{e}}w_{f_{5}}\left\{\Phi^{\prime \prime }-\frac{\eta }{2}Le\left(\zeta -1\right)\Phi^{\prime }+Le\left(\zeta F\Phi^{\prime }+\frac{N_{t}}{LeN_{b}}{\left(\theta^{\prime \prime }\right)}^{2}+\left(\zeta -1\right)\zeta \frac{\partial \Phi }{\partial \zeta }\right)\right\}d\Omega_{e}=0 \end{array}$$(32)$$\begin{array}{c} \int_{\Omega_{e}}w_{f_{6}}\left\{\chi^{\prime \prime }+\frac{Lb}{2}\left(1-\zeta \right)\eta \chi^{\prime }+\zeta LbF\chi^{\prime }-Pe\left(\Phi^{\prime \prime }\left(\delta_{1}+\chi \right)+\chi^{\prime }\Phi^{\prime }\right)-\zeta \left(1-\zeta \right)Lb\frac{\partial \chi }{\partial \zeta }\right\}d\Omega_{e}=0. \end{array}$$

Here $w_{f_{s}}\left(s=1,2,3,4,5,6\right)$ stand for trial functions. Let the domain ($\Omega_{e}$) be divided into 4−nodded elements. The associated approximations of the finite element are:(33)$$\begin{array}{c} F=\sum_{j=1}^{t}\left[\dot{F_{j}}\Upsilon_{j}\left(\zeta ,\eta \right)\right],\;P=\sum_{j=1}^{t}\left[\dot{P_{j}}\Upsilon_{j}\left(\zeta ,\eta \right)\right],\;G=\sum_{j=1}^{t}\left[\dot{G_{j}}\Upsilon_{j}\left(\zeta ,\eta \right)\right],\;\theta =\sum_{j=1}^{t}\left[\dot{\theta_{j}}\Upsilon_{j}\left(\zeta ,\eta \right)\right],\;\Phi =\sum_{j=1}^{t}\left[\dot{\Phi_{j}}\Upsilon_{j}\left(\zeta ,\eta \right)\right]. \end{array}$$
here, $\Upsilon_{j}$ (j = 1,2,3,4) and t=4 For $\Omega_{e}$, the linear-interpolation key functions are defined as follows:.
(34)$$\begin{array}{c} \left.\begin{array}{c} \Upsilon_{1}=\frac{\left(\zeta_{e+1}-\zeta \right)\left(\eta_{e+1}-\eta \right)}{\left(\zeta_{e+1}-\zeta_{e}\right)\left(\eta_{e+1}-\eta_{e}\right)},\;\Upsilon_{2}=\frac{\left(\zeta -\zeta_{e}\right)\left(\eta_{e+1}-\eta \right)}{\left(\zeta_{e+1}-\zeta_{e}\right)\left(\eta_{e+1}-\eta_{e}\right)}\\ \Upsilon_{3}=\frac{\left(\zeta -\zeta_{e}\right)\left(\eta -\eta_{e}\right)}{\left(\zeta_{e+1}-\zeta_{e}\right)\left(\eta_{e+1}-\eta_{e}\right)},\;\Upsilon_{4}=\frac{\left(\zeta_{e+1}-\zeta \right)\left(\eta -\eta_{e}\right)}{\left(\zeta_{e+1}-\zeta_{e}\right)\left(\eta_{e+1}-\eta_{e}\right)} \end{array}\right\} \end{array}$$

Therefore, the stiffness element matrix, matrix of unknowns and the force vector/matrix for the finite element model are followed as: (35)$$\left[\begin{array}{cccccc} \left[L^{11}\right] & \left[L^{12}\right] & \left[L^{13}\right] & \left[L^{14}\right] & \left[L^{15}\right] & \left[L^{16}\right]\\ {}[L^{21}] & \left[L^{22}\right] & \left[L^{23}\right] & \left[L^{24}\right] & \left[L^{25}\right] & \left[L^{26}\right]\\ {}[L^{31}] & \left[L^{32}\right] & \left[L^{33}\right] & \left[L^{34}\right] & \left[L^{35}\right] & \left[L^{36}\right]\\ {}[L^{41}] & \left[L^{42}\right] & \left[L^{43}\right] & \left[L^{44}\right] & \left[L^{45}\right] & \left[L^{46}\right]\\ {}[L^{51}] & \left[L^{52}\right] & \left[L^{53}\right] & \left[L^{54}\right] & \left[L^{55}\right] & \left[L^{56}\right]\\ {}[L^{61}] & \left[L^{62}\right] & \left[L^{63}\right] & \left[L^{64}\right] & \left[L^{65}\right] & \left[L^{66}\right] \end{array}\right]\left[\begin{array}{c} \left\{F\right\}\\ \left\{P\right\}\\ \left\{G\right\}\\ \left\{\theta \right\}\\ \left\{\Phi \right\}\\ \left\{\chi \right\} \end{array}\right]=\left[\begin{array}{c} \left\{R_{1}\right\}\\ \left\{R_{2}\right\}\\ \left\{R_{3}\right\}\\ \left\{R_{4}\right\}\\ \left\{R_{5}\right\}\\ \left\{R_{6}\right\} \end{array}\right]$$
where $\left[L_{mn}\right]$ and $\left[R_{m}\right]$ (*m*, *n* = 1, 2, 3, 4, 5, 6) are expressed as: $$\begin{array}{ccc} L_{ij}^{11} & = & \int_{\Omega_{e}}\Upsilon_{i}\frac{d\Upsilon_{j}}{d\eta }d\Omega_{e},L_{ij}^{12}=-\int_{\Omega_{e}}\Upsilon_{i}\Upsilon_{j}d\Omega_{e},L_{ij}^{13}=L_{ij}^{14}=L_{ij}^{15}=L_{ij}^{21}=L_{ij}^{24}=L_{ij}^{25}=L_{ij}^{26}=0,\\ L_{ij}^{22} & = & -\int_{\Omega_{e}}\frac{d\Upsilon_{i}}{d\eta }\frac{d\Upsilon_{j}}{d\eta }d\Omega_{e}+\frac{1}{2}\left(1-\zeta \right)\eta \int_{\Omega_{e}}\Upsilon_{i}\frac{d\Upsilon_{j}}{d\eta }d\Omega_{e}+\zeta \int_{\Omega_{e}}\bar{F}\Upsilon_{i}\frac{d\Upsilon_{j}}{d\eta }d\Omega_{e}-\zeta \int_{\Omega_{e}}\bar{P}\Upsilon_{i}\Upsilon_{j}d\Omega_{e}-\zeta \left(1-\zeta \right)\int_{\Omega_{e}}\Upsilon_{i}\frac{d\Upsilon_{j}}{d\zeta }d\Omega_{e}\\ & + & 2\beta \zeta \int_{\Omega_{e}}\bar{F}\bar{P}\Upsilon_{i}\frac{d\Upsilon_{j}}{d\eta }d\Omega_{e}+\beta \zeta \int_{\Omega_{e}}{\bar{F}}^{2}\frac{d\Upsilon_{i}}{d\eta }\frac{d\Upsilon_{j}}{d\eta }d\Omega_{e}-M^{2}\zeta \int_{\Omega_{e}}\Upsilon_{i}\Upsilon_{j}d\Omega_{e},\;L_{ij}^{23}=2\lambda \zeta \int_{\Omega_{e}}\Upsilon_{i}\Upsilon_{j}d\Omega_{e}\\ & - & 2\beta \lambda \int_{\Omega_{e}}\zeta \bar{F}\Upsilon_{i}\frac{d\Upsilon_{j}}{d\eta }d\Omega_{e},L_{ij}^{31}=L_{ij}^{34}=L_{ij}^{35}=L_{ij}^{36}=0,\;L_{ij}^{32}=2\lambda \zeta \int_{\Omega_{e}}\Upsilon_{i}\Upsilon_{j}d\Omega_{e}-2\lambda \beta \int_{\Omega_{e}}\zeta \bar{F}\Upsilon_{i}\frac{d\Upsilon_{j}}{d\eta }d\Omega_{e},\\ L_{ij}^{33} & = & -\int_{\Omega_{e}}\frac{d\Upsilon_{i}}{d\eta }\frac{d\Upsilon_{j}}{d\eta }d\Omega_{e}+\frac{1}{2}\left(1-\zeta \right)\eta \int_{\Omega_{e}}\Upsilon_{i}\frac{d\Upsilon_{j}}{d\eta }d\Omega_{e}+\zeta \int_{\Omega_{e}}\bar{F}\Upsilon_{i}\frac{d\Upsilon_{j}}{d\eta }d\Omega_{e}-\zeta \int_{\Omega_{e}}\bar{P}\Upsilon_{i}\Upsilon_{j}d\Omega_{e}-\zeta \left(1-\zeta \right)\int_{\Omega_{e}}\Upsilon_{i}\frac{d\Upsilon_{j}}{d\zeta }d\Omega_{e}\\ & + & 2\beta \zeta \int_{\Omega_{e}}\bar{F}\bar{P}\Upsilon_{i}\frac{d\Upsilon_{j}}{d\eta }d\Omega_{e}+\beta \zeta \int_{\Omega_{e}}{\bar{F}}^{2}\frac{d\Upsilon_{i}}{d\eta }\frac{d\Upsilon_{j}}{d\eta }d\Omega_{e}-2\lambda \beta \int_{\Omega_{e}}\zeta \bar{G}\Upsilon_{i}\Upsilon_{j}d\Omega_{e},L_{ij}^{41}=L_{ij}^{42}=L_{ij}^{43}=0,\\ L_{ij}^{44} & = & -\int_{\Omega_{e}}\frac{d\Upsilon_{i}}{d\eta }\frac{d\Upsilon_{j}}{d\eta }d\Omega_{e}+\frac{Pr}{2}\left(1-\zeta \right)\eta \int_{\Omega_{e}}\Upsilon_{i}\frac{d\Upsilon_{j}}{d\eta }d\Omega_{e}+Pr\zeta \int_{\Omega_{e}}\bar{F}\Upsilon_{i}\frac{d\Upsilon_{j}}{d\eta }d\Omega_{e}+PrN_{b}\int_{\Omega_{e}}{\bar{\Phi }}^{\prime }\Upsilon_{i}\frac{d\Upsilon_{j}}{d\eta }d\Omega_{e}\\ & + & PrN_{t}\int_{\Omega_{e}}{\bar{\theta }}^{\prime }\Upsilon_{i}\frac{d\Upsilon_{j}}{d\eta }d\Omega_{e}-Pr\zeta \left(1-\zeta \right)\int_{\Omega_{e}}\Upsilon_{i}\frac{d\Upsilon_{j}}{d\zeta }d\Omega_{e},L_{ij}^{45}=L_{ij}^{46}=L_{ij}^{51}=L_{ij}^{52}=L_{ij}^{53}=L_{ij}^{56}=0,\\ L_{ij}^{54} & = & -\frac{N_{t}}{N_{b}}\int_{\Omega_{e}}\frac{d\Upsilon_{i}}{d\eta }\frac{d\Upsilon_{j}}{d\eta }d\Omega_{e},L_{ij}^{55}=-\int_{\Omega_{e}}\frac{d\Upsilon_{i}}{d\eta }\frac{d\Upsilon_{j}}{d\eta }d\Omega_{e}+\frac{Le}{2}\left(1-\zeta \right)\eta \int_{\Omega_{e}}\Upsilon_{i}\frac{d\Upsilon_{j}}{d\eta }d\Omega_{e}+Le\zeta \int_{\Omega_{e}}\bar{F}\Upsilon_{i}\frac{d\Upsilon_{j}}{d\eta }d\Omega_{e}\\ & - & Le\zeta \left(1-\zeta \right)\int_{\Omega_{e}}\Upsilon_{i}\frac{d\Upsilon_{j}}{d\zeta }d\Omega_{e},L_{ij}^{61}=L_{ij}^{62}=L_{ij}^{63}=L_{ij}^{64}=0, \end{array}$$
$$\begin{array}{ccc} L_{ij}^{65} & = & -Pe\delta_{1}\int_{\Omega_{e}}\frac{d\Upsilon_{i}}{d\eta }\frac{d\Upsilon_{j}}{d\eta }d\Omega_{e},L_{ij}^{66}=-\int_{\Omega_{e}}\frac{d\Upsilon_{i}}{d\eta }\frac{d\Upsilon_{j}}{d\eta }d\Omega_{e}+\frac{Lb}{2}\left(1-\zeta \right)\eta \int_{\Omega_{e}}\Upsilon_{i}\frac{d\Upsilon_{j}}{d\eta }d\Omega_{e}+Lb\zeta \int_{\Omega_{e}}\bar{F}\Upsilon_{i}\frac{d\Upsilon_{j}}{d\eta }d\Omega_{e}\\ & - & Pe\int_{\Omega_{e}}{\bar{\Phi }}^{\prime }\Upsilon_{i}\frac{d\Upsilon_{j}}{d\eta }d\Omega_{e}-Pe\int_{\Omega_{e}}{\bar{\Phi }}^{\prime \prime }\Upsilon_{i}d\varphi_{j}d\Omega_{e}-Lb\zeta \left(1-\zeta \right)\int_{\Omega_{e}}\Upsilon_{i}\frac{d\Upsilon_{j}}{d\zeta }d\Omega_{e}, \end{array}$$
and
(36)$$\begin{array}{c} R_{i}^{1}=\Gamma ,\;R_{i}^{2}=-\oint_{\Gamma_{e}}\Upsilon_{i}n_{\eta }\frac{\partial P}{\partial \eta }\;ds,\;R_{i}^{3}=-\oint_{\Gamma_{e}}\Upsilon_{i}n_{\eta }\frac{\partial G}{\partial \eta }\;ds,\;R_{i}^{4}=-\oint_{\Gamma_{e}}\Upsilon_{i}n_{\eta }\frac{\partial \theta }{\partial \eta }\;ds,\\ R_{i}^{5}=-\oint_{\Gamma_{e}}\Upsilon_{i}n_{\eta }\frac{\partial \Phi }{\partial \eta }\;ds-\frac{Nt}{Nb}\oint_{\Gamma_{e}}\Upsilon_{i}n_{\eta }\frac{\partial \theta }{\partial \eta }\;ds,\;R_{i}^{6}=-\oint_{\Gamma_{e}}\Upsilon_{i}n_{\eta }\frac{\partial \chi }{\partial \eta }\;ds. \end{array}$$

Here, $\bar{F}=\sum_{j=1}^{t}\left({\bar{F}}_{j}\Upsilon_{j}\right)$, $\bar{G}=\sum_{j=1}^{t}\left({\bar{G}}_{j}\Upsilon_{j}\right)$, $\bar{P}=\sum_{j=1}^{t}\left({\bar{P}}_{j}\Upsilon_{j}\right)$, ${\bar{\theta }}^{\prime }=\sum_{j=1}^{t}\left({\bar{\theta }}_{j}^{\prime }\Upsilon_{j}\right)$, and ${\bar{\Phi }}^{\prime }=\sum_{j=1}^{t}\left({\bar{\Phi }}_{j}^{\prime }\Upsilon_{j}\right)$ are key values that are probably supposed to be renowned. In order to linearize the acquired 61,206 equations with the $10^{-5}$ needed precision, we perform six function evaluations at each node.

## 4. Results and Discussion

This section describes through FE analysis how suction/injection impacts the mechanisms of a Maxwell spinning fluid when it is impacted by the Coriolis effect, magneto-hydrodynamic effects, and micro-organisms. Three different patterns of arcs are mapped on fluctuating values of the intravenous injection/suction (Γ) factor for every figure for these significant quantities, as follows: (Γ=−0.2) (suction), (Γ=0.0) (static), and (Γ=0.2) (injection). The following are the predefined values for the parameters involved: β=0.1 , λ=1.0=M, $N_{b}=0.2=N_{t}$, Le=10, Pr=Lb=5.0, Pe=0.5, $\delta_{1}=0.2$. An analysis of mesh separation is executed to show that the finite element simulations are accurate. The entire zone is split into various grid concentrations of mesh sizes, and there is no further modification after (100×100) has been observed, so all simulations are based on this mesh size (Table 1). For distinctive scenarios, comparisons with previous research are provided in Table 2 and Table 3 to determine the remedy methodology’s accuracy. In certain restrictive instances, it is observed that the existing mathematical evaluations correlate very well with the current investigation. The friction coefficient, as well as the axial and transverse indications $-F^{\prime \prime }\left(0\right),-G\left(0\right)$, are calculated using finite element analysis and are summarized in Table 2 for various values of the rotatory factor (λ)=0.0,1.0,2.0,5.0 when (ζ)=1. The table shows that the computational findings achieved are consistent with the results reported by [52,53]. Furthermore, in Table 3, the Nusselt quantity −θ(0) outputs are consistent with those reported by Bagh et al. [54] and Mustafa et al. [55], who present FEA findings for a variety of values β,λ,Pr, and determine that they are satisfactorily correlated. As a result, certainty in statistical computing is increased, and it is confirmed that the finite element evaluations obtained using the Matlab program show a strong rate of convergence.

### 4.1. Variations of Velocity Profiles

Figure 2, Figure 3, Figure 4 and Figure 5 illustrate the primary and secondary velocity dispersion for various values of the magnetism factor, rotating factor, unsteady factor, and meditation Deborah quantity. Figure 2a,b illustrates the effect of various values of the magnetism factor on the velocity profiles $G^{\prime }\left(\zeta ,\eta \right)$ and $F^{\prime }\left(\zeta ,\eta \right)$. The presence of frictional factors in the context of a Lorentz effect is caused by the incorporation of a stimulating external magnetization and results in the transverse momentum declines shown in Figure 2a, whereas the axial momentum exhibits an inverse relation, as shown in Figure 2b. The axial $F^{\prime }\left(\zeta ,\eta \right)$ and transverse $G^{\prime }\left(\zeta ,\eta \right)$ for various rotating parametric inputs are shown in Figure 3a,b. Figure 3a shows that the Coriolis force causes the transverse momentum to decrease for increasing values of the rotation factor, whereas Figure 3b demonstrates the reverse effect. Figure 4a,b show that the size and thickness of the momentum fluid layers in the axial position increase as the time factor increases, while the viscidity of the momentum fluid layers in the transverse path decreases as the time factor decreases. As a result, it is clear that the unsteadiness factor is crucial for influencing the transverse momentum. Physically, a reduced quantity of fluids is pinched axially with the enhanced viscoelastic effects and fluid is pushed away in a radial direction. Figure 5a,b shows that the Deborah quantity is (β) over the velocity profiles for various values of the tranquility factor. The presence of thermoelastic impacts in the context of delivering the best results in a deflation of the building of transverse momentum is shown in Figure 5a, whereas the tangential momentum exhibits an inverse correlation, as shown in Figure 5b. The increasing relative strength of the rheological effect is associated with a higher meditation quantity, resulting in a decrease in velocity. Additionally, these graphs demonstrate that the F’(ζ,η) profile decreases with increase in Γ=0.2 (injection), but is significantly increased when Γ=−0.2 (suction).

Figure 6a,b shows the graphics of $C_{f_{x}}\sqrt{Re_{x}}$ (friction factor) across the transverse and $C_{f_{y}}\sqrt{Re_{x}}$ (axial direction) closer to the surface for the ζ(0:0.2:1) spectrum and for M(1:1:5). As shown in Figure 6a, increasing ζ(0→1) gradually increases the spread of $\left(C_{f_{x}}\sqrt{Re_{x}}\right)$ until no significant difference is observed. In contrast, the $\left(C_{f_{x}}\sqrt{Re_{x}}\right)$ value adjacent to the plate substrate decreases significantly when *M* is increased. Figure 6b shows that when boosting ζ(0→1), the spread of $\left(C_{f_{x}}\sqrt{Re_{x}}\right)$ changes steadily up to a consistent rate, and then there is no significant variation, whereas *M* continues to increase. A large discrepancy in values adjacent to the surface of the sheet, $\left(C_{f_{y}}\sqrt{Re_{x}}\right)$ can be observed. Physically, the application of a magnetic field normal to the direction of fluid flow gives rise to a force known as Lorentz force. Figure 7a,b shows that the dispersion of $\left(C_{f_{x}}\sqrt{Re_{x}}\right)$ tends to increase at a consistent rate up to a certain point, after which no significant variation for enhancing ζ(0→1) occurs. When λ increases, however, there is a substantial decrease in $\left(C_{f_{x}}\sqrt{Re_{x}}\right)$. When ζ(0→1) is enhanced, the dissemination of $\left(C_{f_{x}}\sqrt{Re_{x}}\right)$ is substantially decreased until no significant change is detected, as shown in Figure 7b, while λ is increased. Furthermore, it is apparent from these infographics that the basic values of $\left(C_{f_{x}}\sqrt{Re_{x}}\right)$ and $\left(C_{f_{y}}\sqrt{Re_{x}}\right)$ for the scenario of Γ=0.2 (injection) are smaller than those for the scenario of Γ=−0.2 (suction).

### 4.2. Temperature Profiles

Figure 8, Figure 9 and Figure 10 illustrate the θ(ζ,η) dissemination when varying the factors involved. The thermal configurations in Figure 8 are enhanced by the magnetic field factor. The cumulative induced resultant force, also known as the resistor Lorentz force, governs the flow momentum between the externally applied magnetic effect and the inner electromagnetic force, as shown in Figure 8a, whereas the wall thickness of the heat transfer performance increases with increasing λ, as shown in Figure 8b. Figure 8a,b show how the thermophoretic factor $\left(N_{t}\right)$ and the Brownian motion factor $\left(N_{b}\right)$ affect the temperature profile. The dispersion of the temperature profile appears to grow as $N_{b}$ and $N_{t}$ inclines. Physically, Nt apply a force on the neighbour particles, the force moving the particles from a hot region to a cold region. Figure 10a,b show the impact of (β) and the time-dependent (τ) on the temperature profile. The tranquility of Deborah’s number and the unsteady factor are enhanced, as are the θ(ζ,η) profiles. Furthermore, it can be seen from these graphs that the temperature decreases with the intensity of Γ=0.2 (injection), while increasing with the Γ=−0.2 (suction) factor. Illustrations of the Nusselt quantity $\left(Nu_{x}{Re_{x}}^{1/2}\right)$ at (0.1:0.1:0.3)Nt&Nb for M&λ are shown in Figure 11a,b. The dispersion of $\left(Nu_{x}\sqrt{Re_{x}}\right)$ decreases subsequently as *M* and λ are accelerated. For increasing values of Nt&Nb, a substantial deterioration in $\left(Nu_{x}\sqrt{Re_{x}}\right)$ occurs close to the panel substrate. Additionally, the figure indicates that for Γ=0.2 (injection) there is a relatively large quantity of $\left(Nu_{x}\sqrt{Re_{x}}\right)$.

### 4.3. Concentration Distributions

Figure 12a,b illustrates that Φ(ζ,η) varies with the magnetic *M*, the rotating factor λ, the Lewis Le, and the Deborah number (β). The concentration profiles are augmented as the magnetic field, rotating field, and relaxation Deborah parameters are enhanced, as illustrated in Figure 12a,b and Figure 13b, respectively. Furthermore, Figure 13a demonstrates that deterioration in the organism’s density increases the Lewis number Le. Physically, a high Lewis number corresponds to a low mass diffusivity, causing the species concentration in the nanofluid to decrease. Figure 14a,b show the progressive behavioural patterns of the local Sherwood number $\left(Shr_{x}\sqrt{Re_{x}}\right)$ at (0.1:0.1:0.3)Nt&Nb for M(0:1:5)&λ(0:1:5). The dispersion of $\left(Shr_{x}\sqrt{Re_{x}}\right)$ is reduced as *M* and λ are increased. In the case of enhancing Nt&Nb, however, a conflicting pattern is observed, and the Γ=0.2 (injection) scenario is higher $\left(Shr_{x}\sqrt{Re_{x}}\right)$ than the Γ=−0.2 (suction) specific case. Figure 15 and Figure 16a,b show (χ(ζ,η)) for variation in *M*, the rotating parameter λ, the bio-convection Lewis number Lb, and the Peclet number (Pe). The microbe dispersion profile is intensified as the magnetic factor *M* and rotation factor λ inputs increase, and it notably tumbles in the context of the bioconvection Lewis number Lb and the Peclet number (Pe) (see Figure 16a,b). Furthermore, it can be seen in the infographics that the microbe dispersion profile χ(ζ,η) decreases when the Γ=0.2 (injection) parameter is used, but it is fractionally increased when the Γ=−0.2 (suction) factor is used. Figure 17a,b shows the trend in the microbe concentration quantity $Re_{x}^{1/2}N_{x}$ for M(0:1:5)&λ(0:1:5) at Nt&Nb(0.1:0.1:0.3), respectively. The $Re_{x}^{1/2}N_{x}$ decreases as *M* and λ increase, whereas the $Re_{x}^{1/2}N_{x}$ increases as Nt&Nb increase. It is also observed that for Γ=0.2 (injection), $Re_{x}^{1/2}N_{x}$ is higher than for Γ=−0.2 (suction).

## 5. Concluding Remarks

In this article, a finite element simulation was exploited to investigate Maxwell nanofluid flows over a bidirectional elongating surface with bio-convection, suction/injection, Coriolis, and Lorentz forces for three-dimensional spinning flow. Based on the results, the following inferences can be made:Increase in the Coriolis and Lorentz’s forces has a decreasing impact on the velocity magnitude, and
has a significant influence on the temperature dispersion and concentration.intensifies the impact of $Cf_{x}Re_{x}^{1/2}$.the Coriolis force causes the transverse momentum to decrease for increasing values of the rotation factor.with the infusion capability, the velocity, temperature, and concentration components are reduced.It is becoming increasingly evident that the simultaneous enhancement of Brownian and thermophoresis factors has a negative effect on the distribution of temperature, and
a declining impact on $Nu_{x}{Re_{x}}^{1/2}$, and positive effects on $Shr_{x}{Re_{x}}^{1/2}$.injection is associated with a larger amount in $Nu_{x}{Re_{x}}^{1/2}$.the injection case has a larger $Shr_{x}{Re_{x}}^{1/2}$ and $Re_{x}^{1/2}N_{x}$ compared to the suction case.the wall thickness of the heat transfer performance increases with increasing rotating parameter.Higher input to the relaxation Deborah number and the unsteady parameter has a negative impact on the magnitude of the primary and secondary velocity, but
has substantial consequences for temperature dispersion.for tiny particles, the volume fraction shows rising effects with higher relaxation Deborah number.Motile microorganism viscosity reduces in the context of augmented bioconvection Peclet and Lewis numbers

This study has involved an analysis of the parameters that their impact on dynamic of fluid flow problems and can be extended in future to include Blasius and Sakiadis flow, and Darcy–Forchheimer and thermoelastic Jeffrey nanofluids.

## Figures and Tables

**Figure 1 nanomaterials-12-03453-f001:**
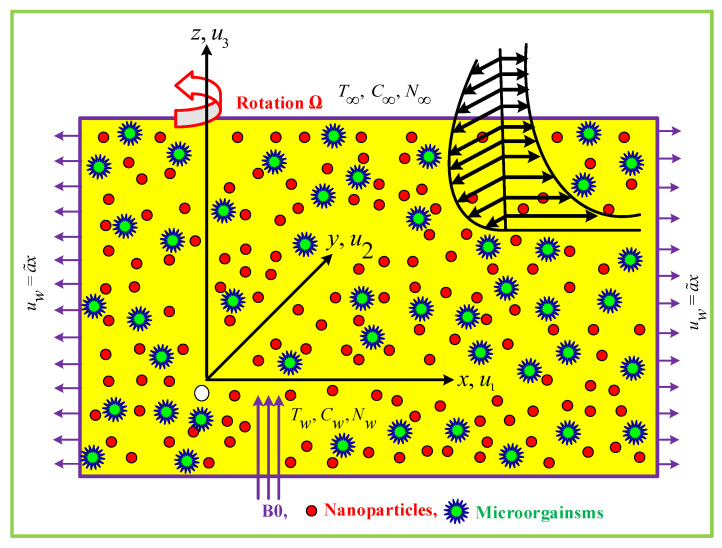
Flow Sketch.

**Figure 2 nanomaterials-12-03453-f002:**
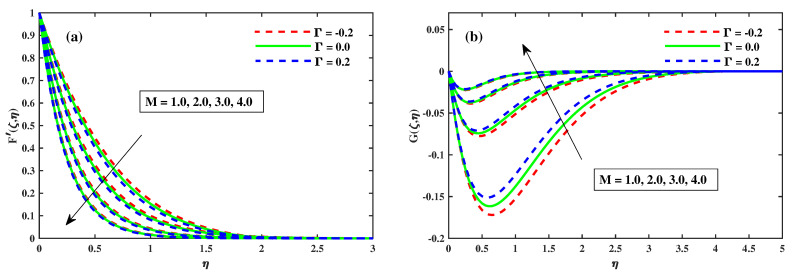
Influence of *M* on *G* along y-direction in (**b**), and $F^{\prime }$ along x-direction in (**a**) when ζ=1.

**Figure 3 nanomaterials-12-03453-f003:**
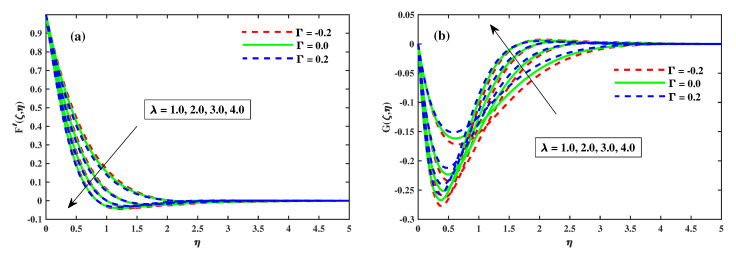
Influence of λ on *G* along y-direction in (**b**) and $F^{\prime }$ along x-direction in (**a**) when ζ=1.

**Figure 4 nanomaterials-12-03453-f004:**
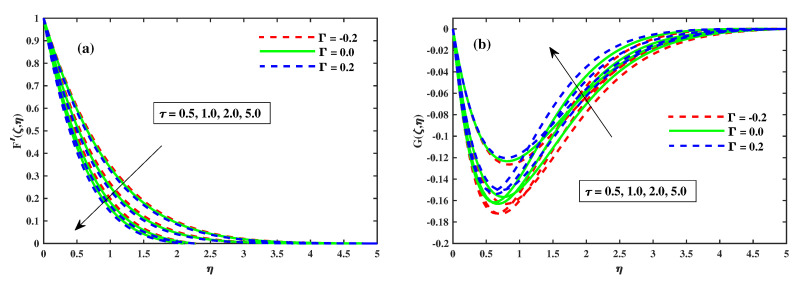
Influence of τ on *G* along y-direction in (**b**) and $F^{\prime }$ along x-direction in (**a**) when ζ=1.

**Figure 5 nanomaterials-12-03453-f005:**
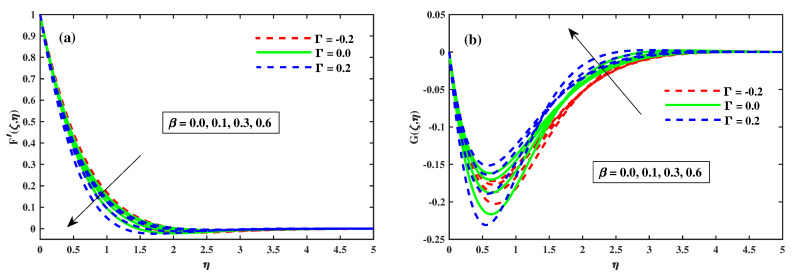
Influence of β on *G* along y-direction in (**b**) and $F^{\prime }$ along x-direction in (**a**) when ζ=1.

**Figure 6 nanomaterials-12-03453-f006:**
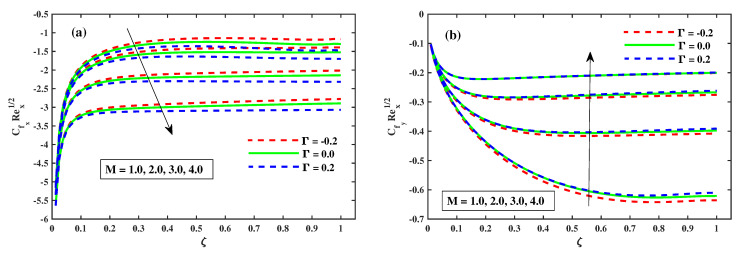
Influence of *M* on $Cf_{x}Re_{x}^{1/2}$ along x-direction in (**a**), and $Cf_{y}Re_{y}^{1/2}$ along y-direction in (**b**).

**Figure 7 nanomaterials-12-03453-f007:**
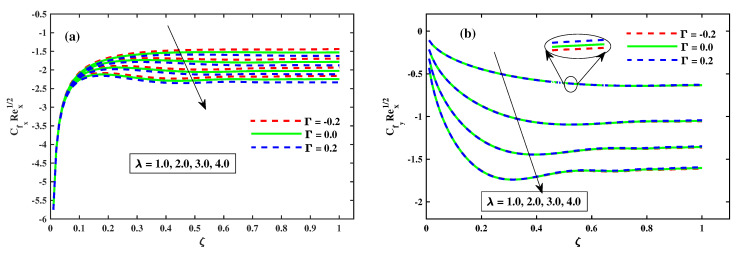
Impact of λ of $Cf_{x}Re_{x}^{1/2}$ along x-direction in (**a**), and $Cf_{y}Re_{y}^{1/2}$ along y-direction in (**b**).

**Figure 8 nanomaterials-12-03453-f008:**
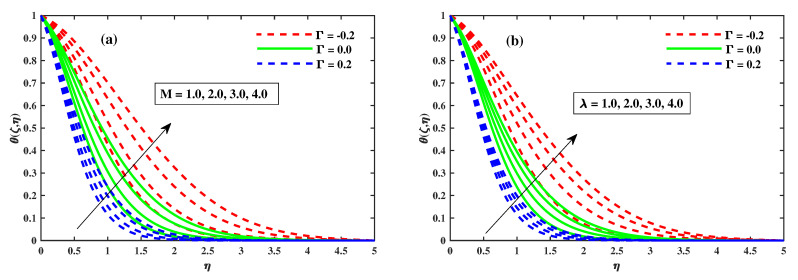
Variation of θ(ζ,η) for *M* in (**a**) and for λ in (**b**) when ζ=1.

**Figure 9 nanomaterials-12-03453-f009:**
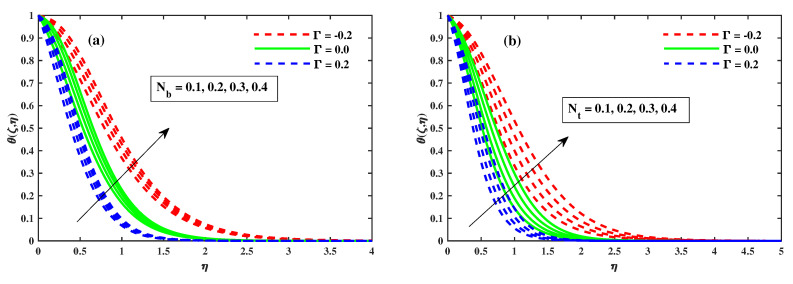
Influence of $N_{b}$ in (**a**) and $N_{t}$ in (**b**) on the temperature profiles.

**Figure 10 nanomaterials-12-03453-f010:**
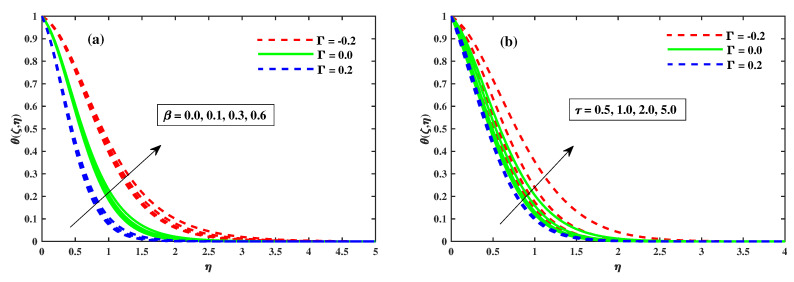
The variation of temperature against β in (**a**) and for τ in (**b**).

**Figure 11 nanomaterials-12-03453-f011:**
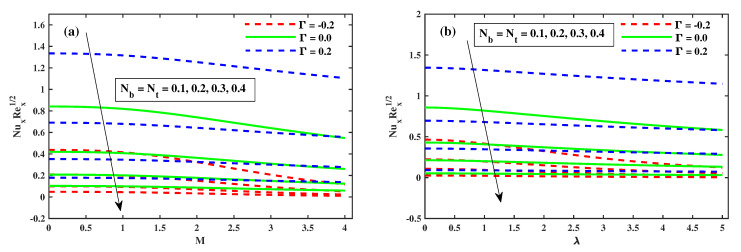
The variation of Nusselt number with $N_{b}$, $N_{t}$ against *M* in (**a**) and against λ in (**b**).

**Figure 12 nanomaterials-12-03453-f012:**
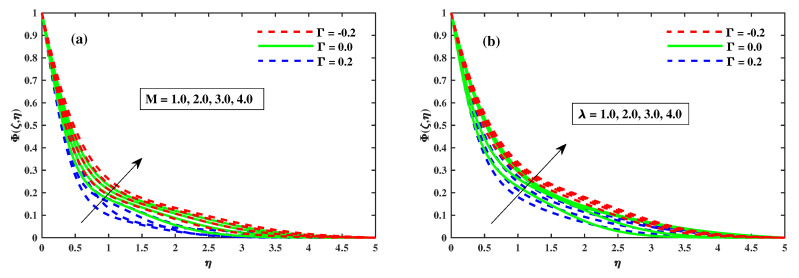
Variation in Φ(ζ,η) for *M* in (**a**) and for λ in (**b**) when ζ=1.

**Figure 13 nanomaterials-12-03453-f013:**
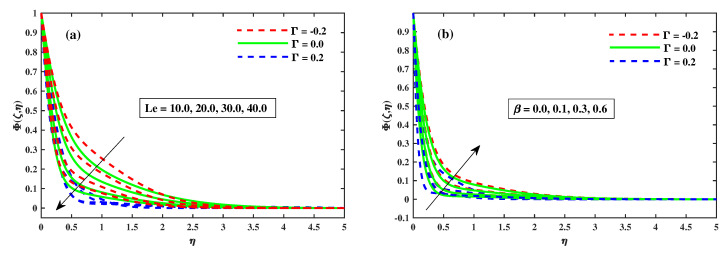
The variation in Φ(ζ,η) against Le in (**a**) and for β in (**b**).

**Figure 14 nanomaterials-12-03453-f014:**
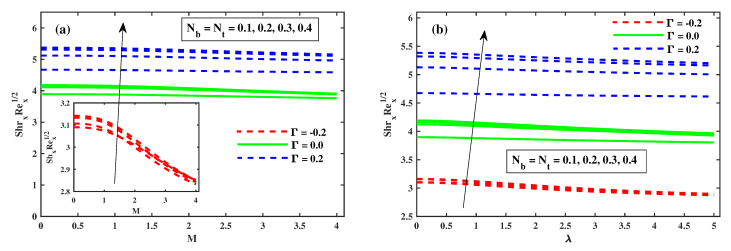
The effect of $N_{b}$, $N_{t}$ for *M* in (**a**) and for λ in (**b**) on Sherwood number.

**Figure 15 nanomaterials-12-03453-f015:**
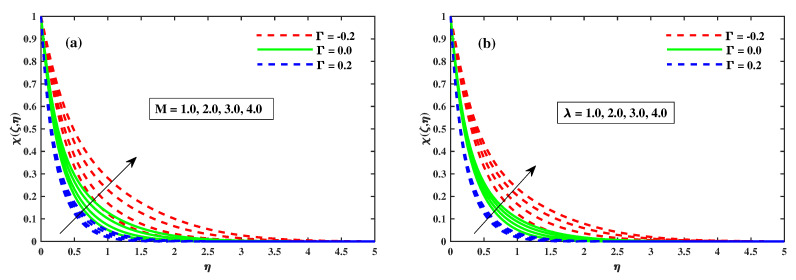
Influence of *M* in (**a**) and λ in (**b**) on χ(ζ,η) when ζ=1.

**Figure 16 nanomaterials-12-03453-f016:**
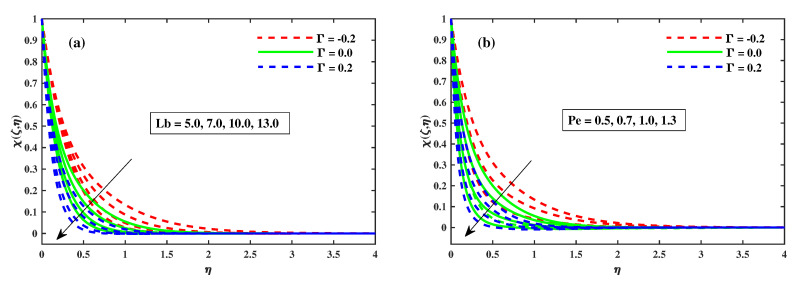
Impact of Lb in (**a**) and Pe in (**b**) on χ(ζ,η) at ζ=1.

**Figure 17 nanomaterials-12-03453-f017:**
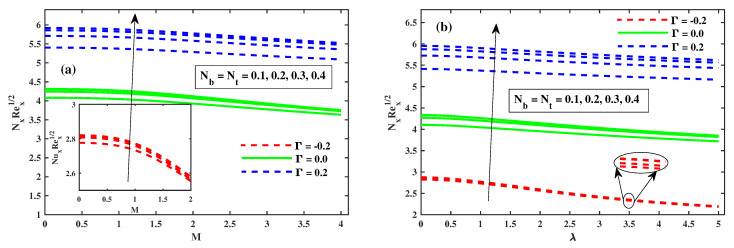
Fluctuation in $N_{x}Re_{x}^{1}/2$ for $N_{b}$, $N_{t}$, along *M* in (**a**) and along λ in (**b**).

**Table 1 nanomaterials-12-03453-t001:** Meshing analysis for various mesh dimensions when ζ=1.0.

Grid Size	$-F^{\prime \prime }\left(\zeta ,0\right)$	$-G^{\prime }\left(\zeta ,0\right)$	$-\theta^{\prime }\left(\zeta ,0\right)$	$-\Phi^{\prime }\left(\zeta ,0\right)$	$-\chi^{\prime }\left(\zeta ,0\right)$
15 × 15	1.7050	0.6876	0.6954	3.7399	4.5771
40 × 40	1.6946	0.6764	0.7399	3.4025	4.8073
70 × 70	1.6935	0.6742	0.7538	3.3371	4.7750
100 × 100	1.6932	0.6736	0.7556	3.3265	4.7565
120 × 120	1.6931	0.6736	0.7558	3.3264	4.7562

**Table 2 nanomaterials-12-03453-t002:** Assessment of $-F^{\prime \prime }\left(0\right)$ and $-G^{\prime }\left(0\right)$ for different values of λ when ζ=1 and other parameters are fixed at zero.

λ	Ali [52]		Wang [53]		Present Results	
	$-F^{\prime \prime }\left(0\right)$	$-G^{\prime }\left(0\right)$	$-F^{\prime \prime }\left(0\right)$	$-G^{\prime }\left(0\right)$	$-F^{\prime \prime }\left(0\right)$	$-G^{\prime }\left(0\right)$
0	01.00000	00.00000	01.0000	00.0000	01.00000	00.00000
1	01.32501	00.83715	01.3250	00.8371	01.32501	00.83715
2	01.65232	01.28732	01.6523	01.2873	01.65232	01.28732
5	02.39026	02.15024	–	–	02.39026	02.15024

**Table 3 nanomaterials-12-03453-t003:** Assessment of $\left\{-\theta^{\prime }\left(0\right)\right\}$ at ζ=1 at various values of Pr, λ and other parameters are fixed at zero.

Pr	β	λ	Ali [54]	Shafique [55]	Present Results (FEM)
1.0	0.20	0.2	00.546683	00.54670	00.5466828
–	0.40	–	00.528090	00.52809	00.5280903
–	0.60	–	–	00.51009	00.5100870
–	0.80	–	00.492547	00.49255	00.4925468

## Data Availability

Not applicable.

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
