# Peer review of "Maxwell Nanofluids: FEM Simulation of the Effects of Suction/Injection on the Dynamics of Rotatory Fluid Subjected to Bioconvection, Lorentz, and Coriolis Forces"

_nanomaterials, 2022, doi:10.3390/nano12193453_

Round 1

Reviewer 1 Report

The work done ‘Maxwell nanofluids: FEM simulation of the effects of suction/injection on the dynamics of rotary fluid subjected to sedimentation, Lorentz, and Coriolis forces’ by respected authors are very much interesting though there are some quarries:

1.       More modern applications of present investigation in the introduction section should be added.

2.      Author should discuss ‘sedimentation, Lorentz, and Coriolis forces’ in introduction section.

3.      Nomenclature with SI units should be mentioned in the revised manuscript.

4.      Magnetic field generally reduces velocity and increase temperature profile of the fluid. But Fig. 8 shows reverse behavior-explain.

5.      Physical significance of every parameter is not included in the discussion section. Skin Friction, Nusselt and Sherwood Number should be explained by engineering pint of view

6.      Revise conclusions as a paragraph containing only significant findings.

7.      The following references have close relevance to the current study, they should be read and cited for the enrichment of the paper:

 https://doi.org/10.1166/jon.2016.1265, https://doi.org/10.1016/j.jppr.2017.01.003, https://doi.org/10.1016/j.powtec.2015.03.043, https://doi.org/10.1166/jon.2021.1812, https://doi.org/10.1080/01430750.2019.1592776, https://doi.org/10.1080/01430750.2021.1955004, https://doi.org/10.1166/jon.2016.1265, https://doi.org/10.1016/j.petrol.2014.12.006, https://doi.org/10.1016/j.jppr.2020.06.002

Author Response

Please find the attached review response

Reviewer 2 Report

Title: 

Maxwell nanofluids: FEM simulation of the effects of suction/injection on the dynamics of rotary fluid subjected to sdimentation, Lorentz, and Coriolis forces?

COMMENTS FOR THE AUTHOR:

Reviewer #: This is a potentially interesting paper that addresses an important TRANSPORT  problem. However, before I can recommend this paper for publication, it should be revised subject to the following suggestions.

  1. Applications of the problem should be included in the abstract.
  2. Pls include units of the symbol.
  3. Definition of Nb, Nt should be interms of alpha (thermal diffusivity) in the denominator. Authors should amend them.
  4. Delete some old reference  and include recent refent refs.

Computation of bio-nano-convection power law slip flow from a needle with blowing effects in a porous medium

Numerical investigation of Von Karman swirling bioconvective nanofluid transport from a rotating disk in a porous medium with Stefan blowing and anisotropic slip effects

Chebyshev collocation computation of magneto-bioconvection nanofluid flow over a wedge with multiple slips and magnetic induction

Magnetohydrodynamic bionanoconvective Naiver slip flow of micropolar fluid in a stretchable horizontal channel

Magnetohydrodynamic bio-nano-convective slip flow with Stefan blowing effects over a rotating disc

Three-dimensional bioconvection nanofluid flow from a bi-axial stretching sheet with anisotropic slip

Comments: minor revision and re-review

Author Response

(The authors gave the same response as above.)

Reviewer 3 Report

Authors have studied the impact of Lorentz and Coriolis forces on the kinetics of gyratory Maxwell Nano fluids flowing against a stretched surface. The gyro-tactic microbes are incorporated to prevent the sedimentation of small particles and to improve consistency. The nanoparticles have been considered due to their captivating properties, which include the ability to enhance thermal dissipation, which is critical in heating systems, advanced technology, microelectronics, and substantial disciplines. The fundamental objective of the analysis has to enhance the heat transformation. An adequate similarity transformations have been used to convert the primary partial differential equations into nonlinear dimensionless ordinary differential equations. The resulting system of equations has been solved using a numerical method termed the Finite Element Analysis. The increasing effects of the Lorentz and Coriolis forces induce the velocities to moderate, whereas the concentration and temperature profiles exhibit the contrary tendency. The intensity, temperature, and velocity variances for the suction scenario are more prominent than those for the injection scenario, but there is an opposite pattern of Nusselt and Sherwood quantities. I recommend this article for publication if authors are adequately address the following comments.

1) There are typos in the manuscript. Correct all. For example, in the title “rotary fluid” I think it is rotatory fluid.  

2) Add nomenclature which contains all the symbols.

3) In the abstract authors have to mention the range of parameters.

4) Results analysis section must be improved by adding physical meaning of influence of parameters.

5) As Maxwell parameter is considered in this analysis, so, in skin friction coefficient formula must contain this parameter. Correct its corresponding formulas and redraw its corresponding graphs.

6) Add the following recent articles to improve introduction part.

Williamson hybrid nanofluid flow over swirling cylinder with Cattaneo–Christov heat flux and gyrotactic microorganism.

Effect of Cattaneo–Christov heat flux on heat and mass transfer characteristics of Maxwell hybrid nanofluid flow over stretching/shrinking sheet.

Combined influence of Brownian motion and thermophoresis on Maxwell three-dimensional nanofluid flow over stretching sheet with chemical reaction and thermal radiation.

Effect of SWCNTs and MWCNTs Maxwell MHD nanofluid flow between two stretchable rotating disks under convective boundary conditions.

Magneto-hydrodynamics heat and mass transfer analysis of single and multi-wall carbon nanotubes over vertical cone with convective boundary condition.

Heat and mass transfer analysis of MWCNTkerosene nanofluid flow over a wedge with thermal radiation.

Author Response

(The authors gave the same response as above.)
